# Development in Fuzzy Logic-Based Rapid Visual Screening Method for Seismic Vulnerability Assessment of Buildings

**Nurullah Bektaş *** and **Orsolya Kegyes-Brassai**

Department of Structural and Geotechnical Engineering, Széchenyi István University, 9026 Győr, Hungary
* Correspondence: nurullahbektas@hotmail.com

**Abstract:** In order to prevent possible loss of life and property, existing building stocks need to be assessed before an impending earthquake. Beyond the examination of large building stocks, rapid evaluation methods are required because the evaluation of even one building utilizing detailed vulnerability assessment methods is computationally expensive. Rapid visual screening (RVS) methods are used to screen and classify existing buildings in large building stocks in earthquake-prone zones prior to or after a catastrophic earthquake. Buildings are assessed using RVS procedures that take into consideration the distinctive features (such as irregularity, construction year, construction quality, and soil type) of each building, which each need to be considered separately. Substantially, studies have been presented to enhance conventional RVS methods in terms of truly identifying building safety levels by using computer algorithms (such as machine learning, fuzzy logic, and neural networks). This study outlines the background research that was conducted in order to establish the parameters for the development of a fuzzy logic-based soft rapid visual screening (S-RVS) method as an alternative to conventional RVS methods. In this investigation, rules, membership functions, transformation values, and defuzzification procedures were established by examining the data of 40 unreinforced masonries (URM) buildings acquired as a consequence of the 2019 Albania earthquake in order to construct a fuzzy logic-based S-RVS method.

**Keywords:** earthquake; seismic assessment; rapid visual screening; fuzzy logic

## 1. Introduction

An earthquake, which is a natural catastrophe, occurs as a result of a significant amount of ground shaking and affects a large area. Even though certain earthquake zones are susceptible to long-period ground motions, the impact of earthquakes that cause significant damage to building stocks is not taken into consideration or given proper concern when structural design projects are prepared and/or buildings are constructed. Additionally, structures in the building stock that are located in seismic-prone areas are vulnerable to hazards if they were constructed prior to the development of seismic design codes. Since determining that possible structural and vital losses could be caused by an impending major earthquake, a pre-earthquake assessment of existing buildings is essential in order to take urgent precautions. Therefore, various seismic vulnerability assessment methodologies are utilized to evaluate existing buildings. These methodologies consist of three stages, starting from rapid visual screening (RVS), preliminary vulnerability assessment (PVA), and detailed vulnerability assessment (DVA).

The structural seismic assessment of buildings may be conducted by using DVA methods. Software utilized for comprehensive structural analysis is employed to develop DVA models. DVA is inherently complex since analyses are conducted while taking into consideration nonlinear structural analysis; the finite element analysis method or the applied element method; and linear or nonlinear material properties. DVA approaches include pushover analysis, time history analysis, incremental dynamic analysis, cloud method, and so on. Using these methods to analyze existing structures is more challenging

than designing a brand-new structure. However, because these techniques are the most reliable, they are used to determine the safety level of certain buildings. Standards such as Eurocode 8 [1], FEMA 356 [2], and FEMA P-695 [3] provide extensive descriptions of these procedures. Employing a high number of design engineers with expertise in structural seismic assessment is required in order to use DVA methods on a large building portfolio. However, since hiring a large number of engineers is costly and these approaches need a lot of computation, less expensive alternatives should be utilized.

Structural material properties, site ground characteristics, and structural drawings must be collected to carry out a PVA method for a second-stage building evaluation approach. To calculate the loads necessary to build even a simple structural model, structured drawings must be collected or developed. After examining existing buildings, if the buildings do not show enough capacity, further DVA methods have to be applied. The data gathered for PVA can also be utilized for DVA.

DVA of building stock is messy; therefore, a reliable RVS method needs to be developed to assess structural vulnerability in a short time compared to PVA and DVA methodologies. Therefore, traditional RVS methods, which were developed for the first stage evaluation of large building stocks, have been utilized. The initial examples of conventional RVS methods are FEMA-154 (ASCE 1988) [4] and FEMA-155 (ASCE 1988) [5]. Subsequently, many other traditional RVS methods, i.e., Europe—EMS-98 Scale [6] and RISK-UE Project [7]; New Zealand—NZSEE [8]; Greece—OASP [9]; Canada—NRC [10]; Italy—GNDT [11]; India—IITK-GSDMA [12]; and Turkey—RBTE-2019 [13] and EMPI [14], have been developed mainly based on the methodology proposed by FEMA 154 and 155 [4,5]. Furthermore, it has been stated by Bektaş and Kegyes-Brassai that some conventional RVS methods were developed for special building types (i.e., hospital buildings and school buildings) [15]. The State Organization of Schools Renovation of Iran (SOSRI) [16], the SAARC Disaster Management Center (SDMC) [17], and the National Institute of Building Sciences (NIBS) [18] all use RVS methods that were developed for pre-earthquake assessment of school buildings. Different RVS techniques have been developed in order to assess various building types. For instance, the Japanese technique can be used to assess reinforced concrete structures while different building types can be assessed using the FEMA P-154 [19], RISK-UE Project [7], and NZSEE [8] methods. Since it is crucial to rank the inspected structures based on their need for repair and further evaluation, buildings can also be categorized according to the requirements for intervention by employing RVS methods.

Although conventional RVS techniques have been widely accepted and tested and they are good at identifying buildings that need to be assessed by DVA methods, they are not good enough to accurately identify building vulnerability class [20]. Therefore, in addition to the numerous nationally developed conventional RVS techniques, there is plenty of other research on the review and comparison [21–23], implementation [24,25], development [26–28], and improvement [29,30] of these techniques. However, according to research conducted by Harirchian and Lahmer [31], the accuracy of the RVS techniques of FEMA P-154 [19] and EMPI [14] is less than 30%. In addition, it is difficult to improve some of the conventional RVS techniques since they were established based on expert opinion (such as NRC [10]). Therefore, by utilizing the knowledge (data) acquired from past earthquakes, soft rapid visual screening (S-RVS) methods have been developed to present more accurate methods in addition to conventional RVS methods. S-RVS methods have been developed in recent years, as they can be easily adjusted taking into account recent advancements and data gathered from previous earthquakes. S-RVS methods are developed using soft computing algorithms, such as machine learning [32–34], fuzzy logic [35,36], and neural networks [37–39].

In one of the initial developments of an S-RVS method, Tesfamariam and Saatcioglu used the fuzzy inference system [40]. This method was developed for reinforced concrete buildings based on building post-earthquake screening data collected after the 1994 Northridge earthquake. Later, Dritsos and Moseley [20] proposed another S-RVS method by combining fuzzy logic and artificial intelligence algorithms. The developed method

demonstrated improvements in terms of accurately classifying the building damage states. It was also highlighted that there is room for further improvements in S-RVS methods. Additionally, fuzzy logic-based S-RVS techniques are often developed for reinforced concrete structures (Harirchian and Lahmer, Moseley and Dritsos, Elwood and Corotis, Demartinos and Dritsos, Moseley and Dritsos, Şen, Ketsap et al. [31,41–46]). Likewise, Mazumder et al. devised a fuzzy logic algorithm-based S-RVS technique for masonry buildings [47].

Since it is crucial to demonstrate the applicability of a developed method, it is necessary to compare its evaluated results with post-earthquake screening data or detailed vulnerability assessment based findings. The developed methods were not always checked using post-earthquake screening data or DVA methods based on evaluation findings [47,48]. Therefore, it is necessary to provide a general development strategy and show the capability of a fuzzy logic-based S-RVS method development to set the relation between building characteristic parameters and vulnerability.

In the last three decades, while developing an accurate RVS method, the implementation of the current conventional RVS techniques resulted in different conservative building vulnerability classification outputs [22,23,31]. Additionally, it is a challenge to draw comparisons even between the revised RVS approach and the prior version in terms of outcomes (such as FEMA 154 [49] and FEMA P-154 [19]). In recent years, in order to develop a future version of conventional methods, soft computing algorithms have been implemented. However, there has not been a study conducted to explain the background research for developing a fuzzy logic-based S-RVS method development. Therefore, this study details the background research acquired to design a fuzzy logic-based S-RVS method, and it defines the parameters that need to be employed as an enhancement to conventional RVS methods. Other studies conducted by authors using post-earthquake building survey data collected after the 2019 Albanian earthquake, which are a continuation of this study, revealed that the proposed method surpasses the conventional RVS method with 57.5 percent [50] and 67.5 percent [51] accuracy, respectively. Finally, this study can be distinguished from previous research by describing crucial calculation steps and reporting the results that demonstrate how an S-RVS technique was established using the recommended methodology.

## 2. Determining RVS Parameters for S-RVS Implementation

To determine the earthquake resistance of buildings, the parameters specific to the buildings and the region need to be considered. As illustrated in Figure 1, some of these parameters include the site's seismicity as shown by the design and site-specific acceleration response spectra [40], the presence of corner columns in URM buildings, and the type of floor (rigid or flexible) [52]. The parameters considered in each of the conventional RVS methods and the effect of each parameter on the computation of the building safety level differ.

By identifying the appropriate parameters employed in the existing conventional RVS methods and organizing the selected parameters in a hierarchical structure, the fuzzy logic-based S-RVS method can be established. A sample hierarchical fuzzy logic-based S-RVS method development that takes into account the parameters of conventional RVS methods is depicted in Figure 2.

In addition to the parameters presented in Figure 2, alterations to the soft story, short columns, infill wall layout, torsional irregularity, pounding possibility, previous damage, maintenance, etc. should be placed suitably in the hierarchical structure. The explanation and examination of some of the parameters given above are explained below.

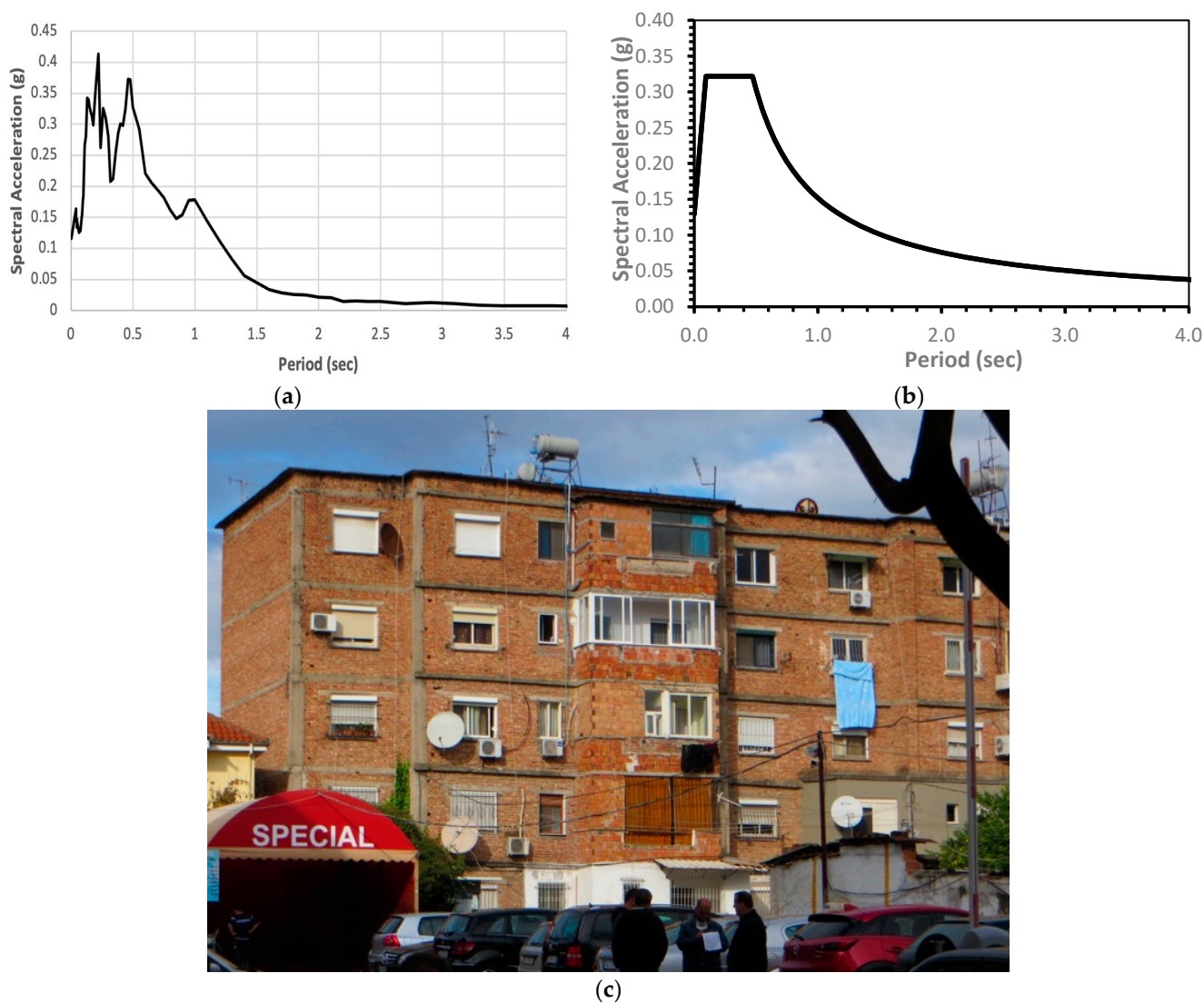

**Figure 1.** Some of the site and building-specific parameters: (**a**) site-specific acceleration response spectrum; (**b**) design acceleration response spectrum; (**c**) URM building with corner columns and rigid floor (Bektaş [50]).

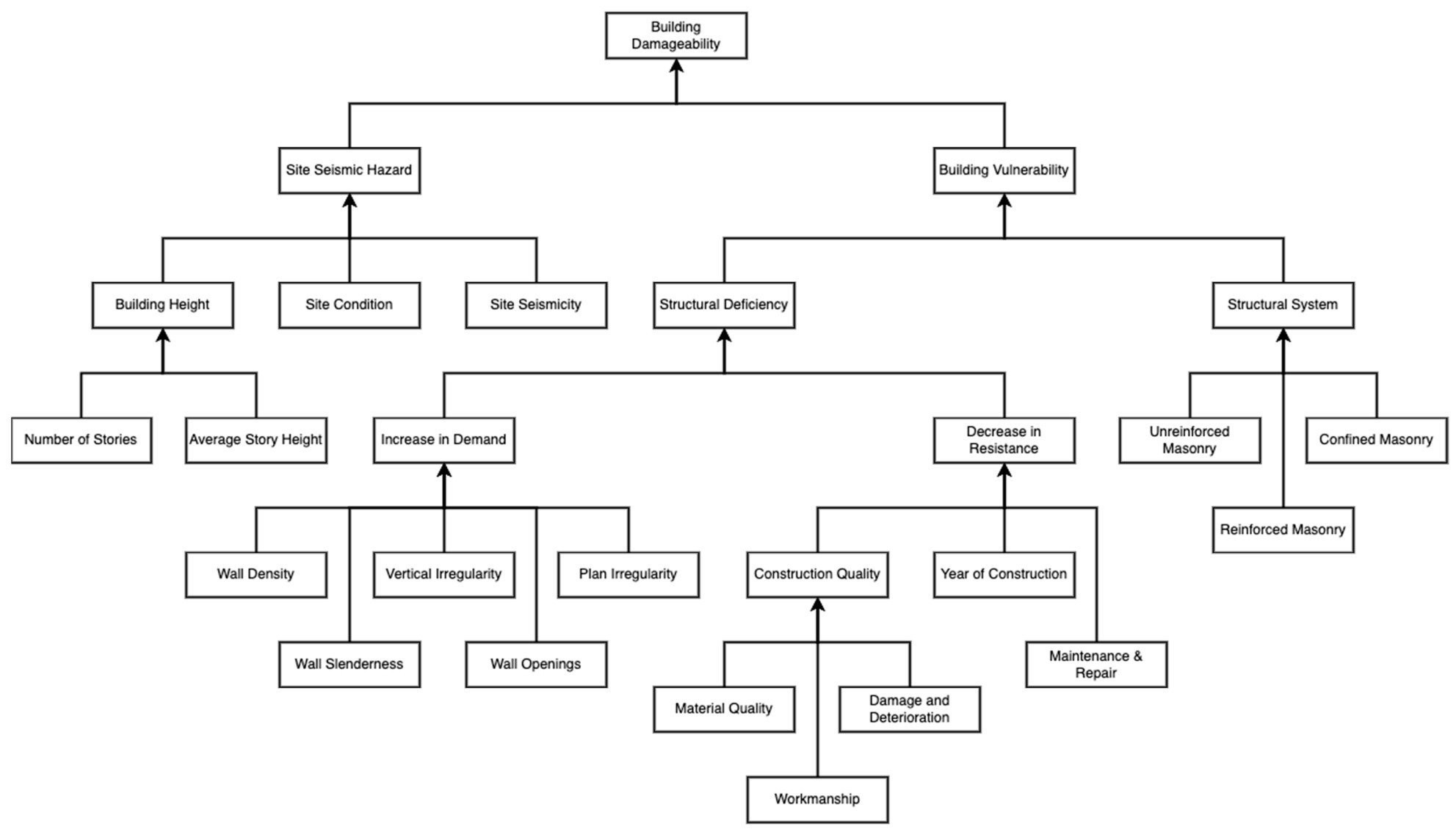

**Figure 2.** Hierarchical fuzzy logic-based S-RVS development flowchart (compiled by authors based on Yadollahi et al., Mazumder et al. & Sivan et al. [31,47,53]).

### 2.1. Plan Irregularity

The structural eccentricity that arises in a structure as a result of an irregularity in the layout is identified as plan irregularity. In earlier earthquakes, plan irregularity has been observed to have a reducing effect on the seismic load resistance of structures by increasing stress concentrations [47]. A structural system that does not have a parallel plan might potentially cause a torsion effect. Torsion is induced by a difference between the center of mass and the center of rigidity caused by structural irregularities in the plan. Large plan and diaphragm openings should be considered critical forms of plan irregularity since they have an adverse effect on seismic load transmission to vertical load-carrying elements. Plan irregularity is a classification of structural plans with re-entrant corners that have plan shapes like E, L, T, U, +, etc. A building is linguistically designated as *Yes* if there are any plan irregularities; otherwise, it is *No*.

### 2.2. Vertical Irregularity

Buildings frequently have vertical irregularities due to usage and architectural concerns. However, building irregularities can have a substantial detrimental impact on the seismic performance of buildings [47]. Among the main factors of damage during strong earthquake excitation, vertical irregularities are considered factors that cause sudden variations in strength and stiffness alongside the height of a building. Vertical irregularities arise when there are discontinuities in the load-carrying system, which include uneven mass distributions at floor levels and changes in the vertical geometric configuration of a building. The location of a building on a steep slope, large wall openings, weak and/or soft stories, variations in floor heights, and in-plane or out-of-plane setbacks are all factors that lead to vertical irregularity. Inconsistencies in the lateral load resisting system (such as weak story, soft story, or setbacks) that cause irregularities to increase the likelihood that buildings would be damaged during an imminent earthquake. Soft and weak stories emerge because of a considerable amount of difference in structural system stiffness and strength variation in the lateral load-carrying system at floor level, respectively. In order to minimize regional stress concentrations, practitioners should follow well-established design procedures and refrain from making dramatic variations to the load-resisting system. The severity of an irregularity in a structure influences the intensity of the corresponding effect of inconsistency. Buildings are linguistically labeled as *Yes* or *No* in conventional RVS methods in terms of vertical irregularity existence.

### 2.3. Construction Quality

The earthquake resilience of existing buildings is degrading, as there is an increased amount of deterioration in material properties. The seismic behavior of structures is influenced by material deterioration, foundation instability, and wall fractures. Furthermore, damages that occur are attributed not only to the deterioration of material attributes but also to poor workmanship. Poor construction quality is caused by poor-quality material, plaster removal, cracks in masonry walls, improper masonry patterns, lack of or non-enforcement of capacity design principles, improper construction procedures, construction errors, and so on. The construction quality of a building is characterized as moderate when there are surface or appearance flaws on floors; ceilings; masonry walls; uneven surfaces; hits; scratches; efflorescence; etc. The construction quality of a building is evaluated as *Poor*, *Moderate*, or *Good* based on all of the factors mentioned above.

### 2.4. Workmanship

Among other parameters, the resilience of a building during an impending earthquake excitation is associated with the quality of workmanship (good execution, faulty execution, etc.) during the construction phase. The application of in-effect official legislation for construction is associated with plumb walls and flat elements as a result of the workmanship. To determine the quality of the workmanship, factors such as concrete quality, construction

quality, vibrator compaction quality, implementation of prepared material at the appropriate time interval after preparation, application of adobe brick construction procedures, and labor mastery are considered. Building quality can be classified as *Poor*, *Average*, or *Good* based on workmanship [54].

### 2.5. Material Quality

The quality of the materials used in building construction influences the capability of the building to withstand severe seismic loads that may arise during an earthquake. Buildings constructed using lower than sufficient-quality materials triggered more severe damage to the structures than envisaged since they lacked the required strength.

### 2.6. Damage and Deterioration

In addition to existing damage to engineering structures, deterioration also adversely affects a building's performance during an earthquake. In this context, in determining a building's seismic performance, existing damage and deterioration in the building are taken into account. Due to a lack of necessary maintenance in an occupied building, deterioration takes place. Some of the deterioration types are time-dependent concrete, mortar, and wall deterioration. Degradations in the structure are also defined by the emergence of cracks or deterioration in elements of buildings. The formation of these sub-parameters may be associated with poor workmanship and/or poor material quality [55]. Based on the existence and/or degree of observed damage and deterioration in a building, the damage and deterioration parameter is classified with linguistic operators of *none*, *minor*, or *severe* [56].

### 2.7. Year of Construction

Since the construction year (age) reflects construction quality, utilized design technique, and the corresponding seismic detailing of members for the building project, it is critical in assessing building vulnerability. The utilized seismic design standard results in the ductility, rigidity, strength, and details of the structure [31]. Due to ancient construction procedures that overlook seismic details in contemporary building requirements, old structures cannot be anticipated to function well during an impending earthquake. The construction year provides information about the building design regulations as well as allows for the classification of corresponding standards of *Low*, *Moderate* and *High*, according to their historical development. For instance, when the development stages of Albanian standards were taken into account, the developed categorization could be calculated, as shown in Table 1.

**Table 1.** Construction year transformation based on the seismic design codes in Albania (compiled by authors based on De Iuliis & Tesfamariam [57,58]).

| Low | Moderate | High | |
|---|---|---|---|
| $YC \leq 1942$ | $1942 \leq YC \leq 1978$ | $1978 \leq YC \leq 1990$ | $YC \geq 1990$ |
| 0.9 | $-0.01 \times YC + 20.27$ | $-0.03 \times YC + 59.8$ | 0.1 |

As a result, in order to adapt the existing classification technique to another location, thresholds need to be altered again based on the seismic standards used in the region under consideration.

### 2.8. Structural System

Different types of structural systems show diverse characteristics in terms of potential damage formation and resistance to damage. Therefore, the type of structural load-carrying system can identify a building's expected lateral strength and ductility during an impending seismic excitation. The seismic vulnerability of unreinforced masonry (URM) and reinforced concrete (RC) structures varies depending on the type of building. Linguistic classes must

be transformed into exact values in order to examine a structural system's effect on building damage states. While URM buildings are classified as *standalone*, *row end*, or *row middle*, RC buildings are classified as C1, C2, or C3, and their transformation is shown in Table 2.

**Table 2.** Structural system transformation for URM and RC buildings (compiled by authors based on Tesfamariam & El Sabbagh [58,59]).

| URM | | | RC | | |
|---|---|---|---|---|---|
| **Standalone** | **Row End** | **Row Middle** | **C1** | **C2** | **C3** |
| 0.9 | 0.85 | 0.8 | 0.7 | 0.25 | 0.35 |

*2.9. Site Seismic Hazard Analysis*

In order to assess earthquake-induced ground motion effects on existing structures, the site seismic hazard module considers the structural period, site seismicity, and site conditions. The seismicity of a region is associated with factors, such as focal depth, rupture type, rupture amount, fault type, and distance from the building site to the fracture site. The impact of seismic activity varies based on the transmission path of the arising seismic energy, site topography, and the characteristics of the bedrock and site soil conditions. The following are steps to determine the site seismic hazard module, as illustrated in Figure 3:

1.  Compute the fundamental structural period ($T_a$) of the building as given in Equation (1).
2.  Evaluate the acceleration response spectra from the site-specific earthquake record as indicated by the blue (continuous) line in Figure 4.
3.  Compute the corresponding spectral acceleration ($S_a$) values indicated by the red scatter to the $T_a$ from the acceleration response spectra shown in Figure 4.

The fundamental structural period ($T_a$) is calculated using the building height, which is evaluated by taking the average floor height and the number of floors into account. In this context, a fundamental structural period ($T_a$) of buildings can be calculated using ASCE/SEI [60] Equation (1).

$$T_a = C_t * h_n^{3/4}$$
$$h_n = n * h_{avg} \tag{1}$$

where, $n$ is the number of floors, $h_{avg}$ is the average story height, $h_n$ is the height of the building, and $C_t$ is the dimensionless coefficient of the fundamental structural period depending on building type (such as $C_t$ = 0.05 for URM buildings and 0.075 for concrete moment resisting frame structures). The site-specific or design acceleration response spectrum was generated to assess corresponding spectral acceleration values to the determined fundamental structural period, as depicted in Figure 4 with red dots. These spectral acceleration values correspond to the fundamental structural periods of two- to nine-story URM buildings from left to right with red dots based on the *x*-axis, respectively. The data on the URM buildings considered in this study were collected after the 2019 Albania earthquake.

Through the application of suitable fuzzy clusters, the determined spectral acceleration values (which are illustrated in Figure 4 with red dots) for the site seismic hazard module were incorporated into the fuzzy logic system-based S-RVS hierarchy shown in Figure 2. Eventually, the site seismic hazard module in conjunction with building vulnerability will be used to determine building damageability.

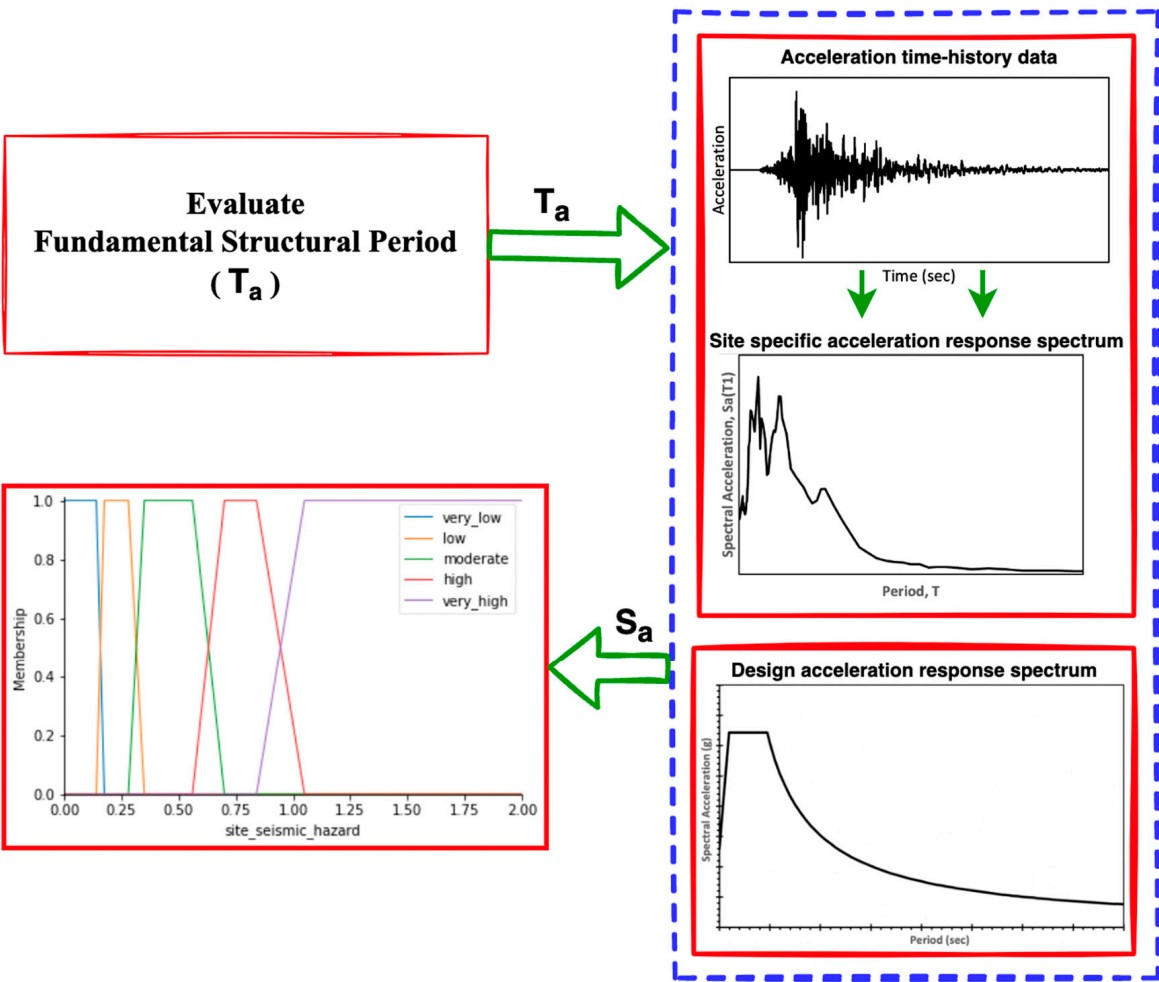

**Figure 3.** Evaluation of site seismic hazard module (compiled by authors based on Tesfamariam & Saatcioglu [40]).

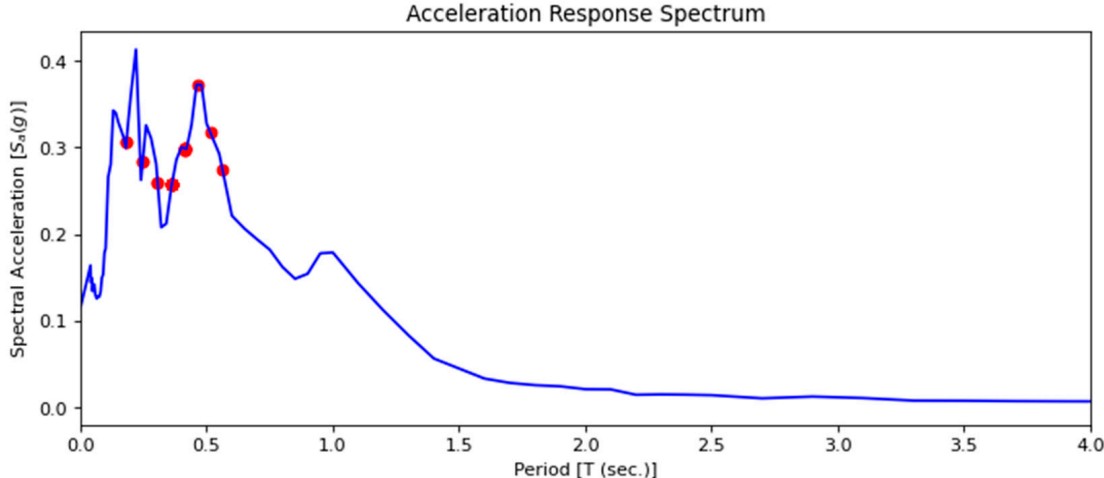

**Figure 4.** Spectral acceleration values for the considered buildings based on building height.

### 2.10. Site and Soil Conditions

Site condition considers the local soil type at the site of the considered building for seismic examination. The local site soil conditions could differ from building to building even in the same neighborhood. Therefore, an in situ investigation needs to be performed to

classify site soil properties. Site characteristics can amplify ground shaking and, on extreme occasions, result in liquefaction and/or landslides. Site seismicity (such as peak ground acceleration in ground surface) is related to the local site soil conditions and subsequently affects the severity of damage in buildings by amplifying earthquake intensity depending on soil layer properties and thicknesses. Site soil properties are classified based on the average shear wave velocity of soil ($V_{s30}$).

### 2.11. Number of Stories

Since higher buildings may undergo greater deformation and cause damage during an earthquake, the building height is an essential performance indicator. In addition, building design regulations restrict the number of floors depending on the seismicity of the area where the building will be built and the type of building. Although the floor numbers are not directly shown with their fuzzy members within the hierarchical schema, the number of floors is used to determine the building height required for the calculation of the site seismic hazard module.

### 2.12. Building Damageability

The damage score assigned to the output variable indicates the likelihood of a structure being exposed to a certain level of damage. The intermediate parameters of building vulnerability and site seismic hazard analysis are used as input parameters to compute building damageability. This integration process yields values ranging from 0 to 1 indicating the degree of building damageability. Building damageability can be divided into different ranges to determine building damage classes. Generally, building damage classes are categorized into five damage levels, such as *very low*, *low*, *moderate*, *high*, and *very high*. The building damageability index on the basis of the considered damage type intervals is linguistically identified in accordance with the computed output. The output index acquired can be translated linguistically to reflect building damageability levels. Building damageability categorization of damage states is displayed in Tables 3 and 4.

**Table 3.** Classification of the building damageability (compiled based on Ploeger [61]).

| None | Light | Moderate | Heavy | Collapse |
|---|---|---|---|---|
| 0.0–0.2 | 0.2–0.4 | 0.4–0.6 | 0.6–0.8 | 0.8–1.0 |

**Table 4.** Description of building damageability intervals (compiled based on ATC-13 [62]).

| **None** | | | | **Destroyed** |
|---|---|---|---|---|
| None | Light | Moderate | Heavy | Collapse |
| 0.0–0.01 | 0.01–0.1 | 0.1–0.3 | 0.3–0.6 | 0.6–1.0 |

## 3. Fuzzy Logic-Based S-RVS Method Development

In 1965, Zadeh [63] proposed fuzzy logic to consider uncertainty and vagueness. Fuzzy logic has the ability to combine qualitative reasoning with numerical calculations. Fuzzy modeling, which establishes the connection between crisp input and crisp output, basically consists of fuzzy sets and fuzzy logic theory. The fuzzy inference system, which is the basic decision-making component of the fuzzy logic system, is divided into three steps: input processing (fuzzification), fuzzy inference engine (rules and inference), and output processing (defuzzification and/or type reduction), as illustrated in Figure 5.

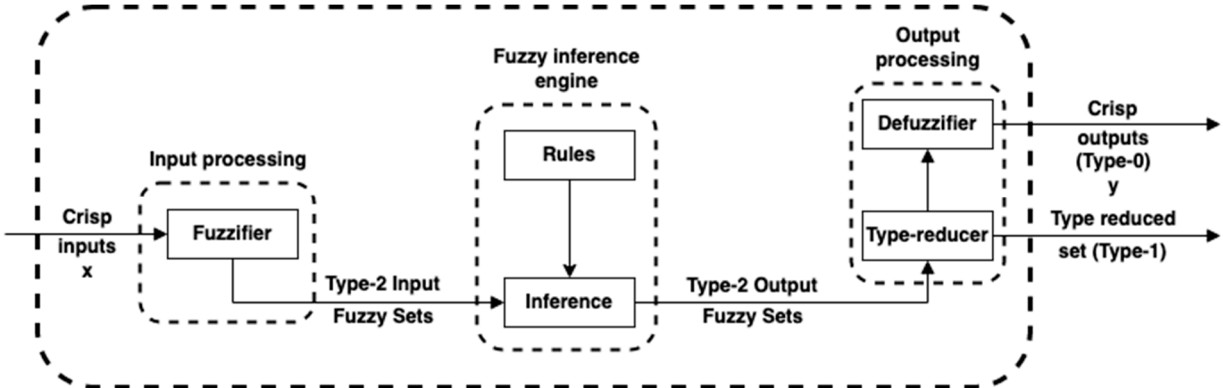

**Figure 5.** Fuzzy logic system.

After completing the determination of parameters, which was presented in the previous section, building screening data needs to be collected to develop S-RVS methods. Then, to enhance existing conventional RVS methods using fuzzy logic or to introduce new S-RVS methods, the necessary definitions for the fuzzy logic implementation schema shown in Figure 4 must be defined. In light of the explanations made in the sub-headings below, the features that need to be considered in order to create the necessary definitions are explained in order. Finally, the essential developments that have been done in this study for fuzzy logic-based S-RVS technique development for URM structures are described in the sub-sections that follow based on data from the 2019 Albania earthquake.

*3.1. Input Processing*

Input processing (granulation or fuzzification) is performed by assigning values between 0 and 1 to linguistic parameters [46]. To quantify linguistic expressions as input points, membership functions with membership values ranging in [0, 1] intervals were employed. In a fuzzy set, 0 denotes non-membership, 1 denotes complete membership in the core of a triangular membership function, and a value between (0, 1) denotes partial membership. Various types of membership functions, such as trapezoidal, triangular, and so on, have been defined in the literature. The membership functions are defined based on expert opinions and previous studies. The context-dependent membership function type was chosen based on usability [64]. Suitable membership functions for some of the input parameters considered in this study were developed by taking into account the knowledge from in-depth literature research and the acquired experience of the authors from reviewing [15] and implementing [65] conventional RVS methods and reviewing and developing S-RVS methods [51,66]. The developed membership functions of the considered input parameters, which are plan irregularity, vertical irregularity, construction quality, construction year, and structural system, are illustrated in Figure 6. Instead of using the linguistic description (*low*, *moderate*, *high*) to represent the intensity of the existence of a parameter, fuzzy logic systems employ transformed values. The *x*-axis of the membership functions represents the existence percentage of the corresponding parameter. Based on the expert's judgment, the range of these values could be decided to be between 0 and 1, or 0 and 100. The corresponding range for the considered membership functions was derived to be between 0 and 1 for the input parameters, as shown in Figure 6. For instance, if the construction quality membership function was taken into consideration, the values of 0 and 1 on the *x*-axis correspond to extremely poor and very good quality observations, respectively. The range of values between 0 (extremely poor) and 1 (very good quality) could also be expressed with linguistic classifications, for example, *low* (*poor*), *moderate* (*moderately poor*), and *high* (*good quality*).

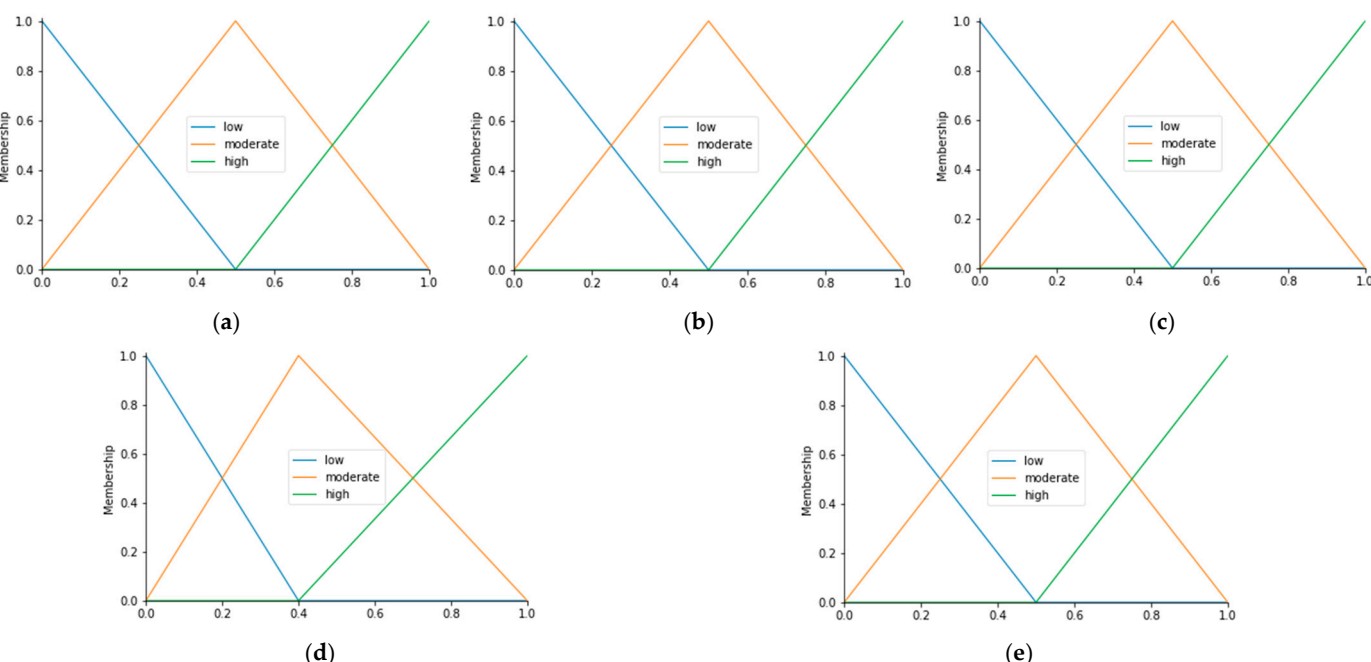

**Figure 6.** Membership functions of input variables: (**a**) plan irregularity; (**b**) vertical irregularity; (**c**) construction quality; (**d**) construction year; and (**e**) structural system.

Intermediate parameters are indicators that have two input variables: an input parameter with an intermediate parameter or two intermediate parameters. Intermediate parameters, in addition to input parameters, are another type of parameter to which membership functions need to be assigned. Intermediate parameters (increased demand, decreased resistance, structural deficiency, and building vulnerability) were obtained as a consequence of processing the input parameters using the fuzzy logic system, which is illustrated in Figure 5. Expert opinion and trial and error-based modification experiments were been performed to develop suitable membership functions for the considered intermediate parameters. The determined values of these intermediate parameters, whose membership functions are illustrated in Figure 7, were employed as input parameters in the subsequent phase, which are presented in Figure 2. Similar to Figure 6, Figure 7 defines the *x*-axis range of the parameters outside the membership function of the site seismic hazard parameter between 0 and 1. However, because the spectral acceleration values of the developed method could be large in extreme events, the *x*-axis of the site seismic hazard membership function was set between 0 and 2 g. In order to evaluate intermediate parameters, the established membership functions of input and intermediate parameters were employed to set up a fuzzy logic system. Then, using the associated crisp input values, intermediate parameters were calculated.

In addition to the input and intermediate parameters considered in this study, the building damageability parameters' membership function needed to be implemented to identify each building's damage state. Building damageability can be defined as the last parameter that is no longer an input parameter. The risk parameters to express building damageability, which were considered in this study to determine the building damage state, are divided into five linguistic categories: *very low*, *low*, *moderate*, *high*, and *very high*. Building damage states were more accurately determined by optimizing the membership functions of the building damageability risk parameters, as shown in Figure 8. The membership function for building damageability in Figure 8 includes an analogous *y*-axis and *x*-axis that, similar to those in Figure 6, exhibit membership values and are a proportion of the parameter that falls between 0 and 1.

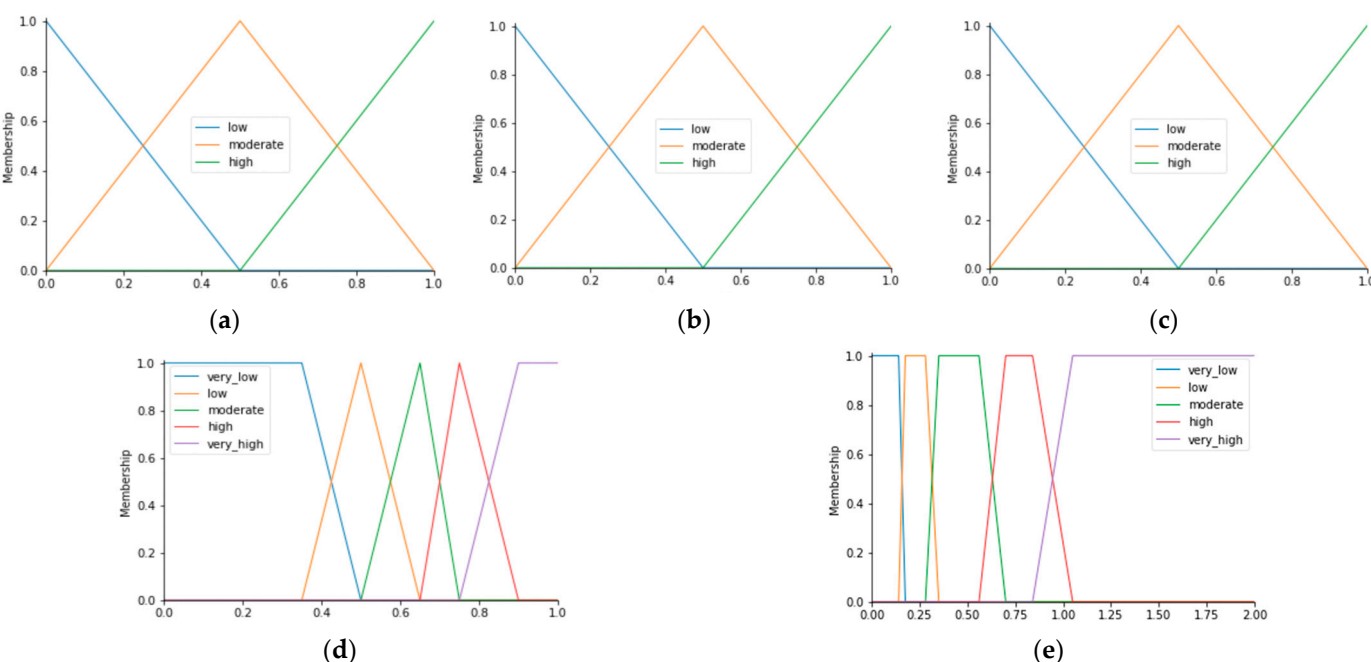

**Figure 7.** Membership functions of intermediate variables: (**a**) increase demand; (**b**) decrease resistance; (**c**) structural deficiency; (**d**) building vulnerability; and (**e**) site seismic hazard.

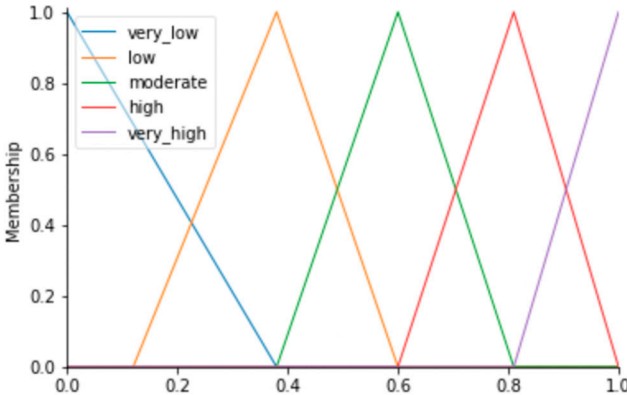

**Figure 8.** Membership functions for building damageability.

Linguistic classifications need to be transformed into numerical values to use linguistically classified building features inside the mathematical calculation algorithm of fuzzy logic. Therefore, defining intervals for parameters of each input characteristic, as employed by Tesfamariam and Saatcioglu [40], to calibrate the model is one of the initial applications of such computations. The calibration of the parameters employed in this investigation [40] was later utilized in other research by Harirchian and Lahmer and Iuliis et al. [31,64]. Based on this calibration, linguistic construction quality categories of *poor*, *moderate*, and *good* were converted into numerical values of 0.99, 0.70, and 0.01 [57,58]. However, it should be noted that these values were calibrated based on the considered post-earthquake screening data of reinforced concrete structures in previous studies. Therefore, it was required to use optimization techniques to transform the linguistic construction quality categories of *poor*, *moderate*, and *good* based on the considered data. In this study, model calibration was conducted by minimizing the root mean square error and increasing the accuracy of the model in terms of determining building damage states based on collected post-earthquake URM building screening data from the 2019 Albania earthquake (Figure 9).

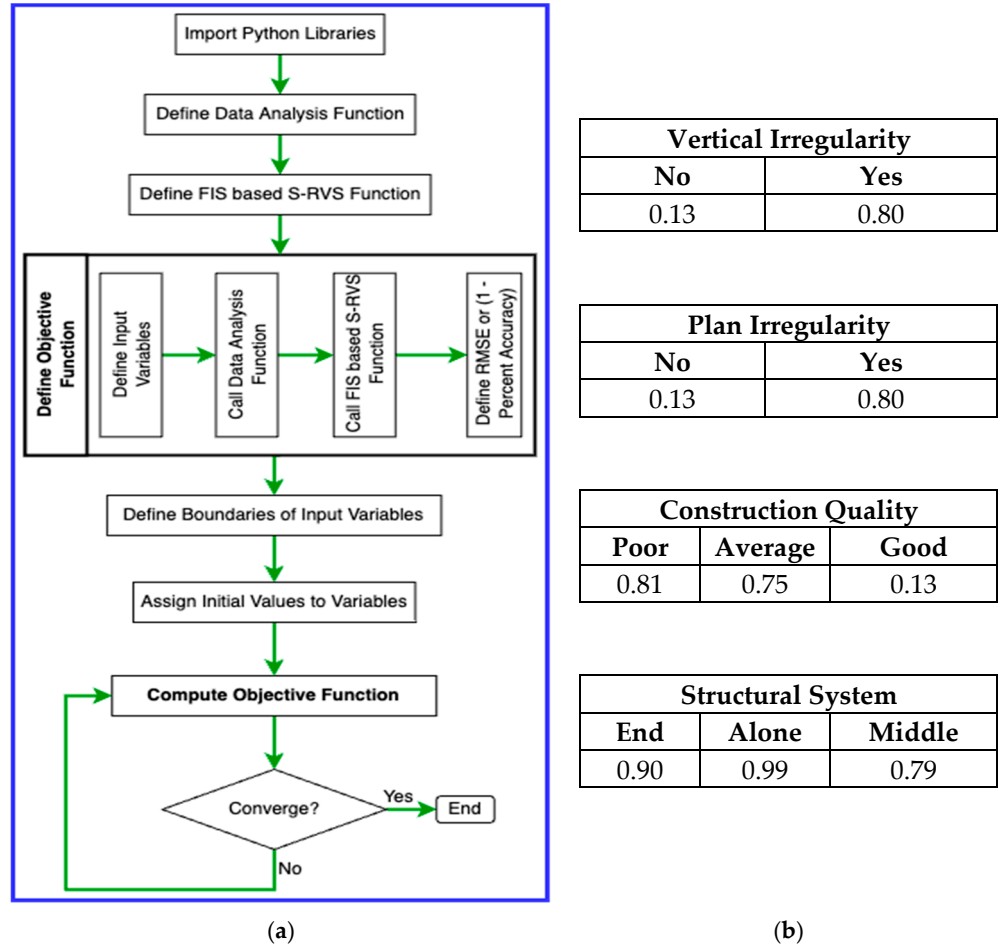

| Vertical Irregularity | |
|---|---|
| **No** | **Yes** |
| 0.13 | 0.80 |

| Plan Irregularity | |
|---|---|
| **No** | **Yes** |
| 0.13 | 0.80 |

| Construction Quality | | |
|---|---|---|
| **Poor** | **Average** | **Good** |
| 0.81 | 0.75 | 0.13 |

| Structural System | | |
|---|---|---|
| **End** | **Alone** | **Middle** |
| 0.90 | 0.99 | 0.79 |

(**a**)                  (**b**)

**Figure 9.** Transformed linguistic parameters (Bektaş et al. [51]): (**a**) flowchart for optimizing transformation values and (**b**) transformed values.

### 3.2. Fuzzy Inference Engine and Rule Formation

A fuzzy inference engine was used to set up a relationship between input and output variables. To characterize the relationship between the premise and consequent values, fuzzy logic-based rules needed to be constructed, which are comprised of expert opinion or previous information. These rule-based statements represent the perspectives of decision makers or judgments concerning an uncertain issue. The if–then Mamdani [67] fuzzy rule base consists of input sets and output based on a conditional statement, as shown in Equation (2).

$$R_i : if \ x_1 \ is \ A_{i1} \ and \ x_2 \ is \ A_{i2} \ then \ y \ is \ B_i, \ \ i = 1, \ 2, \ \ldots, \ n \tag{2}$$

where $x_1$ and $x_2$ denote crisp inputs, $A_{i1}$ and $A_{i2}$ represent the input fuzzy sets (antecedence). Moreover, the fuzzy logic operator *and* is employed in the *if–then* rule to define fuzzy intersection. Additionally, the term *or* is defined as a fuzzy union. $y$ denotes crisp outputs, $B_i$ denotes the output fuzzy sets (consequence), and the number of rules is represented as n. The parameters, which are vertical irregularity, plan irregularity, construction quality, construction year, increase in demand, decrease in resistance, structural deficiency, structural system, site seismic hazard, and building vulnerability, investigated in this study were classified as *Very Low* (*VL*), *Low* (*L*), *Moderate* (*M*), *High* (*H*), and *Very High* (*VH*). Figure 10 presents the generated fuzzy logic-based rule formulation matrices for developing the S-RVS method using some of the widely considered RVS parameters.

| Vertical Irregularity | Plan Irregularity | | |
|---|---|---|---|
| | Low | Moderate | High |
| Low | L | L | M |
| Medium | L | M | H |
| High | M | H | H |

(**a**)

| Construction Quality | Construction Year | | |
|---|---|---|---|
| | Low | Moderate | High |
| Low | L | L | M |
| Medium | L | M | H |
| High | M | H | H |

(**b**)

| Increase in Demand | Decrease in Resistance | | |
|---|---|---|---|
| | Low | Moderate | High |
| Low | L | M | M |
| Medium | L | M | H |
| High | M | H | H |

(**c**)

| Structural Deficiency | Structural System | | |
|---|---|---|---|
| | Low | Moderate | High |
| Low | L | L | M |
| Medium | L | M | H |
| High | M | H | H |

(**d**)

| Site Seismic Hazard | Building Vulnerability | | | | |
|---|---|---|---|---|---|
| | Very Low | Low | Moderate | High | Very High |
| Very Low | VL | VL | VL | L | L |
| Low | VL | VL | L | L | L |
| Moderate | VL | L | L | M | H |
| High | L | L | M | H | VH |
| Very High | L | M | H | VH | VH |

(**e**)

**Figure 10.** Fuzzy rule base matrix representation of some RVS parameters. (**a**) Increase in demand [9 rules]; (**b**) Decrease in resistance [9 rules]; (**c**) Structural deficiency [9 rules]; (**d**) Building vulnerability [9 rules]; (**e**) Building damageability [25 rules].

Finally, the fuzzy inference engine integrated the abovementioned predefined rules and membership functions to conduct the fuzzy logic algorithm for developing an S-RVS method. To provide crisp outputs, the corresponding fuzzy inference engine-based evaluations were defuzzified, as outlined in Section 3.3.

### 3.3. Output Processing and Defuzzification

The output processing stage of fuzzy logic consists of defuzzification or both type-reducer and defuzzification. Defuzzification is the last stage used to convert numerical calculations to crisp output. There are a number of methods that have been developed to perform defuzzification of fuzzy inference engine-based evaluated results. Many defuzzification methods (e.g., the center of gravity, the center of the area, the mean of maxima, and so forth) have been suggested to handle crisp outputs [40,68–70]. The weighted average approach [40,46–48,71,72], centroid [31], graded mean integration technique [73], and center of area method [53] are some defuzzification techniques used to carry out defuzzification process of fuzzy logic-based S-RVS methods. Even though several defuzzification techniques have been developed, they do not always yield the same outcomes [74]. Therefore, the best method, which is highly capable of representing the relation between fuzzy values and crisp outputs, needs to be chosen based on case-by-case circumstances.

Finally, probable input parameters and outcomes will be achieved based on inputs when selecting the defuzzification technique that needs to be considered. Some of the key aspects taken into account when deciding on the defuzzification technique are extensively detailed below using building vulnerability and site seismic hazard analysis as the input parameters for building damageability. For instance, when both the site seismic hazard analysis and the building vulnerability are zero (not existence), the building damageability is anticipated to be zero as well. Alternately, when building vulnerability and site seismic

hazard analysis are one (full existence), building damageability is anticipated to be one. The above-given defuzzification methods were employed sequentially in the developed S-RVS method. Since the largest of the maxima defuzzification method incorporates the criteria described above, it was employed in the development of the S-RVS method. Furthermore, the implementation of the proper calibration techniques and considerations are extensively illustrated in Section 3.4.

### 3.4. Model Calibration

Rules, membership functions, and transformation values may all be adjusted to conduct model calibration for the development of an accurate fuzzy inference system-based S-RVS technique. In addition, the membership functions and rules could be adjusted based on expert opinion, optimization, or post-earthquake screening data employed in the adaptive neuro-fuzzy inference system (ANFIS) framework [75]. The optimization of input variables was used to conduct the transformation of linguistic variables [58], so instead of the existence of vertical irregularity being defined as *yes*, it was transformed to 0.8. Nevertheless, boundaries for each variable and objective function needed to be established for this implementation in the Python programming language [76] based on written code, as shown in Figure 9a. The SciPy library of the Python programming language was utilized to calculate transformation values. Data from post-earthquake building screenings were imported into the objective function, and relevant data analysis was conducted to prepare the data for fuzzy logic system implementation. The optimization system was then prepared to operate after integrating defined variables into the data frame and specifying an objective function (minimizing either root mean square error or 1 minus one-to-one damage state accuracy percentage).

The optimum transformation values are shown in Figure 9b based on the use of the optimization approach, which is depicted in Figure 9a. Eventually, it should be highlighted that the generated model may be capable of having more than one set of appropriate optimized transformation values. Therefore, the best candidate set should be considered.

### 3.5. Validation and Comparison of the Developed Method

Since the accuracy of findings obtained from RVS methods is not expected to be very high [77], more accurate building assessment data (such as post-earthquake screening and/or detailed vulnerability assessment-based building examination data) needs to be used to demonstrate the applicability of the developed S-RVS methods. Post-earthquake building screening data was typically utilized to validate proposed RVS approaches in the literature [68,78] and by the first author of study [50]. However, the suggested technique may also be adjusted in accordance with the information gleaned through detailed building vulnerability assessments. In addition, the information retrieved from the building examination, subject to expert judgment, can also be utilized to calibrate the proposed method.

The percentage of the buildings allocated to each damage state by employing an RVS method was compared with real-world post-seismic damage state percentages of buildings. Although this approach shows an important correlation among the number of buildings assigned in each damage state, it is not sufficient by itself to describe the accuracy of the system. Therefore, one-to-one, one class more severe, and one class less severe classification of damage states were also considered to demonstrate the capability of the system to identify building damage states, as was used by [50,51]. One-to-one building damage state classification accuracy of the system was defined by comparing real damage states and determined damage states, as performed by [50]. If the accuracy of the one-to-one classification of an evenly distributed data-based method was high, it directly demonstrated the high reliability of the proposed system. Although the one-to-one building damage states classification is of utmost importance, it was necessary to carefully examine the classification at a level up and down as a result of evaluation-based categorization, as performed by [51]. When buildings with different characteristics than the buildings considered for developing the S-RVS method were utilized, it was vital to

demonstrate if the method tends to classify building damage states as one step less or more severe before taking necessary precautions. Last but not least, the proportion or quantity of building categories could be represented visually in a format that is simple to comprehend and interpret, such as a confusion matrix as shown in Figure 11. In addition to showing one-to-one building damage classification in diagonal cells, confusion matrices also illustrate how much the damage states of buildings are classified into less severe and more severe categories.

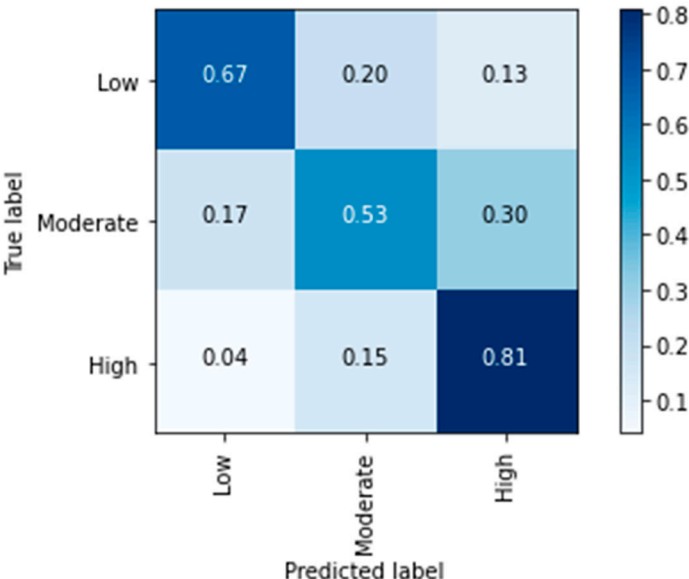

**Figure 11.** A representative sample confusion matrix.

## 4. A Representative Case Study

In 2019, an earthquake measuring 6.4-moment magnitude occurred in Durrës, Albania. The hypocenter of the earthquake is 22 km in depth. There was extensive damage to nearby residential locations. After the earthquake, the screening data of 40 URM buildings were gathered by a team dispatched by Hungarian authorities. Figure 12 depicts representative pictures that reflect the distinguishing aspects of the buildings under consideration. The story levels of these buildings were rigid slabs, which were constructed from reinforced concrete slabs.

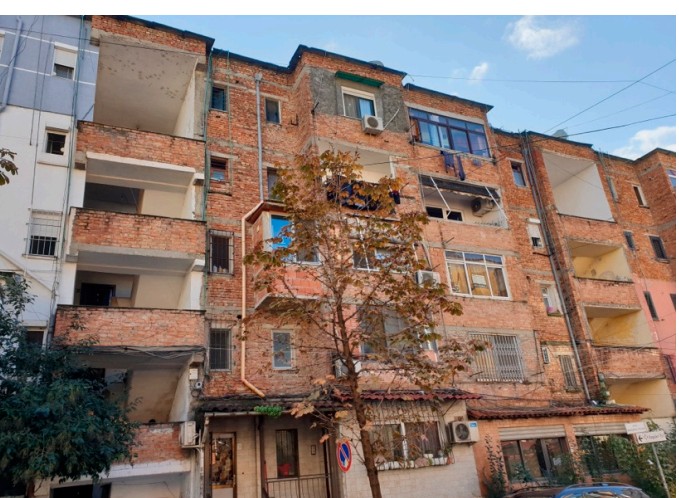 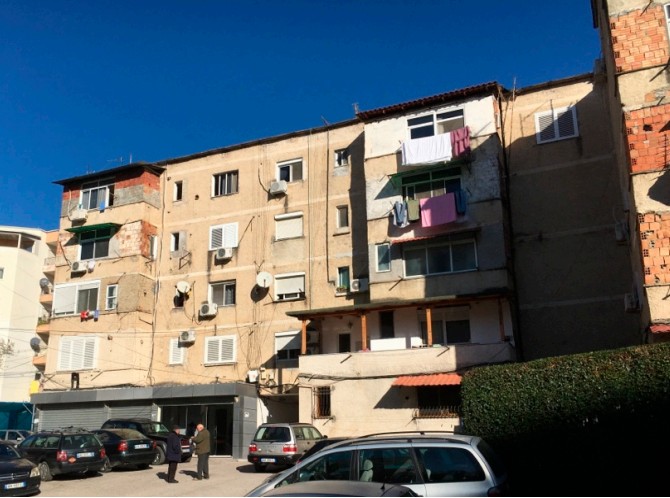

**Figure 12.** Representative pictures of the building screening data collected after the 2019 Albania earthquake.

Table 5 illustrates the sample building screening data used to verify the S-RVS technique that was established. A total of 15% of the examined buildings had a *high (D3)* damage class, 20% had a *moderate (D2)* damage class, and 65% had a *low (D1)* damage class. Bektaş et al. [51] used building evaluation data together with the above-described approach to develop a fuzzy logic-based S-RVS method.

**Table 5.** 2019 Albania post-earthquake building screening data to validate the S-RVS method.

| Building ID | Vertical Irregularity | Plan Irregularity | Construction Quality | Construction Year | Structural System | Number of Floors | Damage State |
|---|---|---|---|---|---|---|---|
| 1 | No | No | Poor | Before 1942 | End | 5 | D3 |
| 2 | Yes | No | Poor | Before 1942 | Middle | 3 | D2 |
| 3 | Yes | Yes | Moderate | Before 1942 | Alone | 5 | D3 |
| 4 | Yes | Yes | Good | 1962–1963 | End | 4 | D3 |
| 5 | No | No | Poor | 1975 | Middle | 5 | D3 |
| 6 | No | No | Good | Before 1942 | End | 2 | D3 |
| 7 | Yes | Yes | Moderate | Before 1942 | Alone | 3 | D2 |
| 8 | Yes | No | Poor | 1965–1972 | Middle | 5 | D2 |
| 9 | No | Yes | Poor | 1984 | End | 5 | D2 |
| 10 | No | No | Moderate | Before 1942 | End | 2 | D2 |
| 11 | No | Yes | Poor | Before 1942 | End | 5 | D3 |
| 12 | Yes | Yes | Good | Before 1942 | End | 6 | D1 |
| 13 | Yes | No | Good | Before 1942 | Alone | 6 | D1 |
| 14 | No | No | Good | Before 1942 | End | 5 | D2 |
| 15 | Yes | No | Good | Before 1942 | Alone | 5 | D1 |

It is challenging to establish a straightforward correlation between building damage state and building characteristic parameters. Therefore, to understand the building damage state relationship with some of the taken-into-account elements (vertical and plan irregularity, construction quality, structural system), Figure 13 was drawn. While the y-axes of Figure 13 represent the percentages in terms of damage states, the x-axes represent the linguistic classes of the considered parameters, such as *yes* or *no* for vertical and plan irregularity. Most buildings with and without vertical and plan irregularities were classified in the damage state as *low*, as shown in Figure 13a,b. The building damage class was generally classified as *low* when the construction quality was *good* or *moderate*, as shown in Figure 13c. However, it is challenging to draw this distinction for *poor* quality buildings. As can be seen in Figure 13d, 85% of the *alone* buildings were classified in damage state *low*, and 59% of *end* buildings were classified in damage state *low*; however, *middle* buildings were mostly classified in *low* and *high* damage states.

Finally, a fuzzy logic-based S-RVS method was developed utilizing the building screening data of URM structures that were gathered following the 2019 earthquake in Albania. The following section provides explanations of relevant findings and discussions.

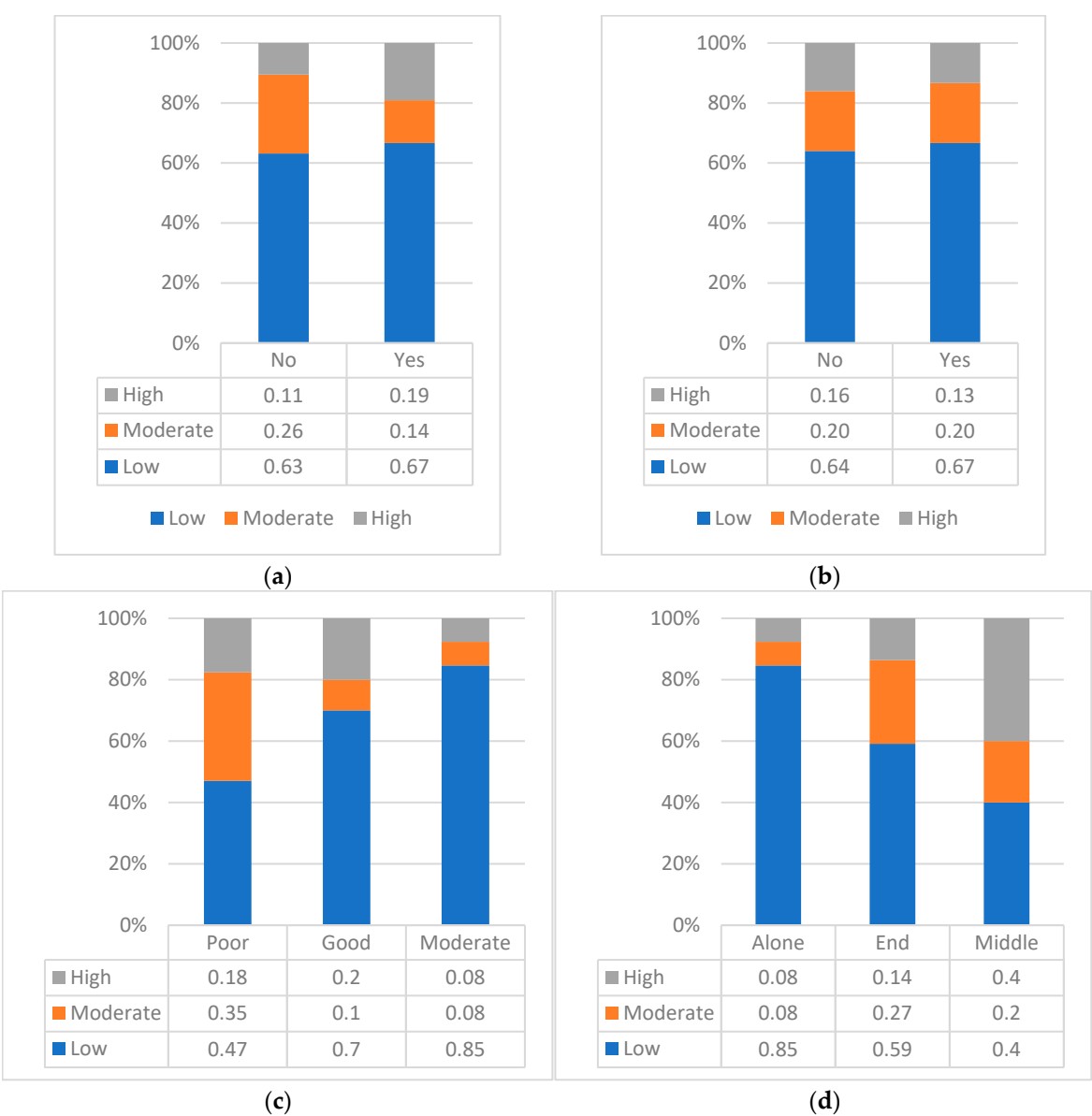

**Figure 13.** Building damage state relation with (**a**) vertical irregularity, (**b**) plan irregularity, (**c**) construction quality, and (**d**) structural system.

## 5. Results and Discussion

Although there are several papers on the development of fuzzy logic-based S-RVS methods by different researchers [20,47,53], the necessary steps and parameters for the development of a fuzzy logic-based S-RVS method have not been presented previously. This paper offers details about the presented S-RVS methodology. Subsequently, the advancements of the S-RVS method developed by the authors in another study [51]—a continuation of this study—compared to the conventional RVS methods are also presented here.

Figure 2 illustrates a generic hierarchical structure composed of a variety of parameters; the developed S-RVS method by the authors considers an optimized number of parameters. The Albanian post-earthquake (2019) building assessment data were used for this study. The second section of this study provides a comprehensive overview for establishing the parameters taken into account. The development of an S-RVS technique based on a fuzzy inference system was described in Section 3. In Section 4, which summarizes the case study, all the information presented prior to implementation was employed to develop the S-RVS method.

For the development of an effective S-RVS method, the selection of an appropriate defuzzification method and accompanying calibration technique was highly significant. Transformation values, rules, and membership functions could be defined purely based on expert opinion; optimization techniques were implemented to calibrate the developed S-RVS method.

Even though there are many studies conducted to develop S-RVS methods by employing fuzzy logic algorithms, these methods did not consider zero (not existence) or one (full existence) of all input parameters in the hierarchical S-RVS system. When we consider the existence of zero or one of all input parameters, building damageability is also required to give zero or one value. Eventually, the provided S-RVS method met the aforementioned conditions.

Additionally, several defuzzification techniques were used to select the best method. Therefore, this issue and the corresponding defuzzification techniques were presented in Section 3.3. The largest of maxima was ultimately chosen as the most effective defuzzification method for this investigation.

This article outlines the background research that contributed to the development of the fuzzy logic-based S-RVS method. The application process and related results are based on data collected from the 2019 Albania earthquake, which were presented by the authors in another paper [51]. The developed method's accuracy of 67.5 percent was shown by the use of the techniques described in Section 3.5. The accuracy of the developed S-RVS method significantly outperforms the accuracy rate of the prior study (57.5%) conducted to develop a fuzzy logic-based S-RVS method by the first author [50].

Limited research in the literature has examined the accuracy of established fuzzy logic-based S-RVS methods by using post-earthquake screening data or DVA methods. Harirchian and Lahmer [31] and Bektaş [50] contrasted the outcomes of implementing the designed S-RVS technique and demonstrated the correctness rate of the method. Table 6 provides a comparison of the developed S-RVS methods in which accuracy rates are demonstrated by comparing the post-earthquake screening data with the method presented in this study based on the suggested methodology.

**Table 6.** Comparison of the developed S-RVS method with the literature.

|  | **Bektaş [50]** | | **Bektaş et al. [51]** | | **Harirchian and Lahmer [31]** | |
|---|---|---|---|---|---|---|
| Building type/Accuracy | URM | 57.5% | URM | 67.5% | RC | 62.2% |

Given that conventional RVS techniques (FEMA P-154 [19] and EMPI [14]) have been shown to have less than 30% accuracy in correctly identifying building damage states [31], it can be concluded that the suggested methodology-based, established S-RVS technique is evidently more accurate than conventional RVS methods. As a consequence, it has been shown that the technique developed utilizing the methodology presented in this research is highly accurate when compared to both earlier S-RVS methods and traditional RVS methods. The outcomes not only demonstrate the method's applicability but also its capability for future advancements.

As a further development, a neural networks (NNs) algorithm can be interconnected with the fuzzy logic algorithm to enhance the S-RVS method by taking into account the site specificity of post-earthquake building screenings or detailed vulnerability assessments based on the collected data. To perform such interconnection-based analyses, the ANFIS [75] environment, which is available as a Python library and within the Matlab fuzzy logic toolbox, can be utilized to calibrate rules, membership functions, and transformation values. Even though a fuzzy logic system is a static system (meaning that the amount of data has no impact on the accuracy of the determination), the ANFIS environment is utilized to transform the developed fuzzy logic-based S-RVS method into a dynamic system. In addition, the ANFIS environment enables the S-RVS method to have self-enhancement capabilities, such as machine learning and NNs. The ANFIS environment will be employed

in future investigations to improve the developed S-RVS method. Finally, when employing NNs instead of developing an S-RVS method using a pure fuzzy logic algorithm, the distribution and quantity of data employed are crucial factors. For this reason, further uniformly dispersed data need to be gathered to develop an S-RVS method based on linked NNs with fuzzy logic algorithms.

Finally, to demonstrate the accuracy of the developed method, evenly distributed post-earthquake screening data and/or detailed vulnerability assessment based on collected building assessment data is required, which is the intention of the authors in further studies.

## 6. Conclusions

In order to assess potential losses before an impending earthquake or to gauge the extent of damage after one, RVS methods are used to examine building stocks, which is more time-effective than other methods, e.g., DVA. Based on the experience that the authors have gathered from employing conventional RVS methods, these methods have shown less accuracy compared to field data, so it is necessary to enhance the reliability of these methods. In order to aid future researchers, the development of a fuzzy logic-based S-RVS method was completely addressed and discussed in this paper. Therefore, the classification of building damage states before and after an earthquake might be performed more accurately using the system that has been proposed by the authors.

The parameters of conventional RVS methods were used by the developed and presented S-RVS method, which is shown in Figure 2. The screener may quickly gather some of the corresponding input parameters that were taken into consideration (such as vertical irregularity, plan irregularity, and structural system). Even if it is challenging to establish a straightforward link between input parameters and the state of building damage, the established S-RVS method is capable of forming a relationship between the parameters taken into consideration. The developed method showed an accuracy rate of 67.5 percent, significantly higher than existing RVS methods [51].

Given the subjectivity of field experts performing field screening and the ambiguity of parameters, it is advised to apply a method based on fuzzy logic, such as the S-RVS method, which was explained in this paper and the continuation study [51] of the authors.

**Author Contributions:** Conceptualization, N.B. and O.K.-B.; methodology, N.B.; software, N.B.; validation, N.B.; formal analysis, N.B.; investigation, N.B.; resources, N.B. and O.K.-B.; data curation, N.B.; writing—original draft preparation, N.B.; writing—review and editing, N.B. and O.K.-B.; visualization, N.B.; supervision, O.K.-B. All authors have read and agreed to the published version of the manuscript.

**Funding:** This research received no external funding.

**Institutional Review Board Statement:** Not applicable.

**Informed Consent Statement:** Not applicable.

**Data Availability Statement:** Not applicable.

**Conflicts of Interest:** The authors declare no conflict of interest.

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
