# Peer review of "Development in Fuzzy Logic-Based Rapid Visual Screening Method for Seismic Vulnerability Assessment of Buildings"

_geosciences, doi:10.3390/geosciences13010006_

Round 1

Reviewer 1 Report

This paper reports the development of a methodology based on a rapid visual screening with the aim to evaluate the seismic vulnerability of buildings.
As a case study, the authors use some data from Albania post-earthquake 2019.

As a general comment, the study is interesting but it needs some integrations and improvements related to: the state of the art, the related works, the description of the case study and the methodology.

Form te reviewer's perspective:
- it is suggested a revision of the introduction in order to clearly identify i)what is the "problem", ii) are there any existing solutions in literature, iii) what gap do you whant to fill.

- as reported in the second section: "To determine the earthquake resistance of buildings, the parameters specific to the buildings and the region are required to be considered". In addition to the description, some images/diagrams could improve the impact of the paper. Moreover, the effect of these parameters on the earthquake resistance of the buildings needs to appropriate references.

- the results need some additional comments

Author Response

Firstly, we would like to express our gratitude to Reviewer – 1 for his/her time and significant feedback. The modifications and/or additions made in response to the given comments below are explained one by one.

This paper reports the development of a methodology based on a rapid visual screening with the aim to evaluate the seismic vulnerability of buildings.

As a case study, the authors use some data from Albania post-earthquake 2019.

As a general comment, the study is interesting but it needs some integrations and improvements related to: the state of the art, the related works, the description of the case study and the methodology.

Besides the explanations added for the clarification of the state of the art and the related work in the Introduction (Pages from 2 to 3), another section was added named A Representative Case Study. This section explains the refereed case study and the considered post-earthquake building screening data. (Pages from 16 to 18)

Form the reviewer's perspective:

  • it is suggested a revision of the introduction in order to clearly identify

As the reviewer suggested introduction section was extensively revised. The corresponding modifications are explained below under the related commend. (Pages from 2 to 3)

  1. what is the "problem",

The problem with conventional RVS methods is explained following part of introduction.

  • The sentence between lines 79 to 81
  • The sentences between lines 84 to 87
  • The sentences between lines 111 to 115

The problem with fuzzy logic based developed S-RVS methods is explained following part of the introduction.

  • The sentence between lines 106 to 108
  • The sentence between lines 116 to 118
  1. are there any existing solutions in literature,

The sentences between lines 90 and 93 illustrate existing solutions by developing S-RVS methods.

The sentence that begins at line 94 summarizes some of the developed  S-RVS methods based on fuzzy logic.

  • what gap do you want to fill.

The following points in the manuscript indicate the gap that this study seeks to fill. (Page 3)

  • Sentence in lines 108 to 110
  • Lines from 116 to 120
  • Sentence start from lines 123 to 126
  • as reported in the second section: "To determine the earthquake resistance of buildings, the parameters specific to the buildings and the region are required to be considered". In addition to the description, some images/diagrams could improve the impact of the paper. Moreover, the effect of these parameters on the earthquake resistance of the buildings needs to appropriate references.

A new figure was added to enhance the explanation by graphically displaying it, as the reviewer noted. (Pages from 3 to 4 – Figure 1) The new figure was explained with a new phrase and appropriate reference. (Lines from 129 to 132)

  • the results need some additional comments

The Results and Discussion section is extended by comparing the developed S-RVS method with the first author's previous method and another fuzzy logic-based developed S-RVS method from literature in the paragraph, which starts from line 575 – page 19. The corresponding comparison is illustrated in Table 7. (Page 19)

Besides, the findings of the developed method’s result compared with the accuracy rates of the conventional RVS methods in the paragraph, which starts from line 584. (Page 19)

The adjustments and additions made in this study are shown in green text in the paper, making it easy to follow all of the changes.

Reviewer 2 Report

The manuscript has 19 pages, 52 references, 9 figures, 5 tables. The manuscript is challenging to establish a straightforward link between an input parameter and the state of the building damage, the established S-RVS method is capable of forming a relationship between the parameters taken into consideration. Albanian post-earthquake (2019) building assessment data is used for this study. However, I don’t see information about this earthquake (parameters or damage structure).

There are explanations about fuzzy logic based rapid visual screening method for seismic vulnerability assessment of buildings using Membership functions of input variables and Intermediate parameters. The fuzzy inference system, which is the basic decision making component of the fuzzy logic system, is divided into three steps: input processing (fuzzification), fuzzy inference engine (rules and inference), and output processing (defuzzification and/or type reduction). All steps are clearly presented with the values in linkage matrix.

Comments:

Authors’ affiliation is the same. Not necessary to repeat it by different numbers.  

The Geoscience MDPI template is not used here. Authors need to construct manuscript with the template. Sections need to be numbered. So phases from p. 15-16 like “under the title …” would be shorter.

Reference style is not appropriate. (Manuscript accepted.) or in print?

[11] in Italian? In Spanish?

[13] Ministry for Environment and Urban Planning, … (2019). Which Country “Ministry of Turkey”? [4] Country, Publisher city?

Authors always refer to the reference [19], when postulate “building screening data collected after the 2019 Albania earthquake, conducted by the authors revealed that the proposed method surpasses the conventional RVS method with an accuracy of 67.5 percent”. Unfortunately as I see (Manuscript in preparation.). This is not right to affirm something, not visible original results for the readers.

Turkey – RTBE-2019 [13]. Are you sure about this model name?

Some repetitions are annoying, when they repeat at the close sentences: e.g. p. 2 many traditional RVS methods established. Based on … limitation of traditional RVS methods. p. 10 10

membership functions … were determined as triangular membership functions.

p. 4 starts with the parameters description: 1. Plan Irregularity 2. Vertical Irregularity … I suggest to number them alphabetically (a) (b) as in Figure 5, where they are mentioned.

Figure 2 bad quality.

Figure 3 – title should be moved to the figure place.

P. 9. Section numbering? Anew 1?

P. 10. I didn’t get how to get Intermediate parameters. There is no explanation about (a) Increase demand (b) Decrease resistance (c) Structural deficiency.

P. 11 before 2. Section “earthquake, as illustrated in Figure 8” the figure should be place right here.

Author Response

Initially, we would like to thank the Reviewer – 2 for his/her time and significant comments. Changes and/or additions made in the light of the comments made are explained separately under each of the comments listed below.

The manuscript has 19 pages, 52 references, 9 figures, 5 tables. The manuscript is challenging to establish a straightforward link between an input parameter and the state of the building damage, the established S-RVS method is capable of forming a relationship between the parameters taken into consideration. Albanian post-earthquake (2019) building assessment data is used for this study. However, I don’t see information about this earthquake (parameters or damage structure).

It is challenging to make a connection between the input parameters and the building damage state, as the reviewer noted. Another part, A Representative Case Study, is introduced to further demystify these difficulties for the reader. Under the newly added section, more description of the 2019 Albania earthquake and the data under consideration is presented.

There are explanations about fuzzy logic based rapid visual screening method for seismic vulnerability assessment of buildings using Membership functions of input variables and Intermediate parameters. The fuzzy inference system, which is the basic decision making component of the fuzzy logic system, is divided into three steps: input processing (fuzzification), fuzzy inference engine (rules and inference), and output processing (defuzzification and/or type reduction). All steps are clearly presented with the values in linkage matrix.

Comments:

  1. Some repetitions are annoying, when they repeat at the close sentences: e.g.
    1. 2 many traditional RVS methods established. Based on … limitation of traditional RVS methods.
  • The highlighted statement by the reviewer has been removed, and additional explanations have been added to provide a more thorough understanding of the topic.
    1. 10 10 membership functions … were determined as triangular membership functions.
  • The last sentence of the first paragraph under the heading Input Processing has been altered, as the reviewer noted. (line 349)
  1. Authors always refer to the reference [19], when postulate “building screening data collected after the 2019 Albania earthquake, conducted by the authors revealed that the proposed method surpasses the conventional RVS method with an accuracy of 67.5 percent”. Unfortunately, as I see (Manuscript in preparation.). This is not right to affirm something, not visible original results for the readers.

We would like to highlight that the Sustainability journal is reviewing the cited manuscript. The corresponding citation was changed.

  1. The Geoscience MDPI template is not used here. Authors need to construct manuscript with the template. Sections need to be numbered. So, phases from p. 15-16 like “under the title …” would be shorter.
  • The template of the manuscript changed to MDPI template. Sections were numbered as given in the manuscript.
  • The sentence at the end of the second paragraph of the Results and Discussion section had rewritten to shorten the explanation. (Line 547)
  • Another sentence at the end of third paragraph of the Results and Discussion section had been shortened by removing “as expressed under the title of Model Calibration.” (Line 556)
  • Another sentence at the 5th paragraph of the Results and Discussion section had been rewritten. (Line 564)
  1. Authors’ affiliation is the same. Not necessary to repeat it by different numbers.

Authors’ affiliation has been modified as stated by the reviewer.

  1. Reference style is not appropriate. (Manuscript accepted.) or in print?
  • [11] in Italian? In Spanish?
  • [13] Ministry for Environment and Urban Planning, … (2019). Which Country “Ministry of Turkey”? [4] Country, Publisher city?

Corresponding references were modified as requested by the reviewer.

  1. Turkey – RTBE-2019 [13]. Are you sure about this model name?

Instead of RTBE, the model named is RBTE. The corresponding corrections had been performed. Similar usage of the model name (RBTE) can be found in [1,2]. (Line 67)

  1. 4 starts with the parameters description: 1. Plan Irregularity 2. Vertical Irregularity … I suggest to number them alphabetically (a) (b) as in Figure 5, where they are mentioned.

Instead of naming as a, b, c, …, the numbering suggested by the MDPI template like 2.1, 2.2, ..., 2.12, had been performed for section names. Even though the authors want to modify the bullet style for subheadings as stated by the reviewer, the MDPI format had been used. However, the numbering of figures is changed based on the titles’ numbers as 1, 2, 3, 7, 8. (Page 11)

  1. Figure 2 bad quality.

The corresponding figure has been modified to enhance visual quality, as requested by the reviewer. (Page 8–9)

  1. Figure 3 – title should be moved to the figure place.

As stated by the reviewer, the title is moved to the figure place and the Figure was renamed. (Page 9)

  1. 9. Section numbering? Anew 1?

As stated by the reviewer section numbering had been performed based on the MDPI format.

  1. 10. I didn’t get how to get Intermediate parameters. There is no explanation about
  2. Increase demand
  3. Decrease resistance
  4. Structural deficiency.

As mentioned by the review, two new sentences have been added to the end of the second paragraph of the Input Processing section to explain the evaluation of intermediate parameters in detail. In addition, further modifications performed in this paragraph to make the necessary explanations are illustrated with green text. (Lines 353 to 366)

  1. 11 before 2. Section “earthquake, as illustrated in Figure 8” the figure should be place right here.

The corresponding figure  was moved to end of the section 3.1 as recommended by the reviewer. (Page 13)

The adjustments and additions made in this study are shown in green text in the paper, making it easy to follow all of the changes.

References:

[1]       T.P. DoÄŸan, T. Kızılkula, M. Mohammadi, İ.H. Erkan, H. Tekeli KabaÅŸ, M.H. Arslan, A comparative study on the rapid seismic evaluation methods of reinforced concrete buildings, International Journal of Disaster Risk Reduction. 56 (2021) 102143. https://doi.org/10.1016/j.ijdrr.2021.102143.

[2]       Y.L. TezÄ¡, E. Gülgeç, Betonarme Yapıların Deprem Performanslarının Belirlenmesi Için Kullanılan Hızlı DeÄŸerlendirme Metotlarının KarşılaÅŸtırılması, 2019.

Round 2

Reviewer 1 Report

Dear authors, thank you for your response. My comments have been considered and required revision has been performed.

Author Response

Dear reviewer, I want to inform you that we will make a request to our institution for an additional linguistic check of the manuscript. We would like to express our gratitude for accepting our modifications!

Reviewer 2 Report

The manuscript looks better with the mdpi template, but still is hard to read for non-specialists. 

Comments:

Check Affiliations and write appropriate - work+address

Line 55 PVA method. Abbreviation. Need to be described before as full name.

Line 120. Authors still refer to their previous work (which is still under review). It seems to me not so proven. At least it should be" accepted" or "in print". Plus, the journal's title is important. Don't the authors really have any other results (conference, workshop) to cite here? 

Line 131. "URM buildings" should be explained. What is it? Maybe refer to the Fig. 1.

I do not like subsection 2.9. It has uncertain places. Line 257 "Error! Reference source not found". Lines 260&262 not Figure 4, but Figure 3! Line 257 not just period, but "fundamental structural period". Please, try to save figure titles NEAR the figure.

OK, we see an equation in the figure 3. So, the repetition in the text is excessive. Line 279 what is "FIS system", which was never mention before?

Figure 4. Spectral acceleration values ... corresponding to the building height. Where is height? I see only period. What red dots mean?

Finally, I am lost. Authors call Ta with different names. Line 265 - building fundamental period ; Line 267 - approximate fundamental period; line 271 -  fundamental building period; Figure 3 - fundamental structural period. Please, use the same term in the entire text.

Ct dimension? If Ta in period is sec, h - height in meters. But upon Fig. 3 - the coefficient of building period is equal 0.05.

Line 315 style for "Table 4."

Line 395 too early to Figure 9. The mention in the text is in line 458.

Figure 1 and 11 copyright? by authors?

Figure 5. upon Zadeh [63]?

Figure 7d. d) Building vulnerability (e) Site seismic hazard. In the figure, I observe "very low low moderate high very_high". However, only in Table 5 (below) I see, that only Building vulnerability and Site seismic hazard have 5 classes, and other - 3 classes. This needs more in the subsections of Section 2. Also are you sure "2.12. Building Damageability", so where do you explain  Building vulnerability ?

detailed vulnerability assessment (DVA) is it a method or conception for complex analysis? 

Tables: inappropriate style. Please, refer to the table template.

All figures move right. 

In my opinion, authors need to acknowledge anonymous reviewers, as they find so many small mistakes. And reviewers help significantly in manuscript improving. 

References.

First of all, journal's titles - italic!

For books and guidance - Publisher city and country,+Publisher. or online link. 

e.g. [10-14] - "." at the end not ";"

[67] initials! Mamdani E.H. 

Author Response

Firstly, we would like to express our gratitude to Reviewer – 2 for his/her time and significant feedback. The modifications and/or additions made in response to the given comments below are explained one by one.

The manuscript looks better with the mdpi template, but still is hard to read for non-specialists.

Comments:

  1. Check Affiliations and write appropriate - work+address

The affiliations are modified as stated by the reviewer.

  1. Line 55 PVA method. Abbreviation. Need to be described before as full name.

The corresponding abbreviation is already explained in line 36.

  1. detailed vulnerability assessment (DVA) is it a method or conception for complex analysis?

Detailed vulnerability assessment (DVA) is a methodology used for the vulnerability assessment of buildings. DVA methodology and corresponding methods are briefly described in lines from 34 to 37 and the second paragraph of the introduction.

  1. Line 120. Authors still refer to their previous work (which is still under review). It seems to me not so proven. At least it should be" accepted" or "in print". Plus, the journal's title is important. Don't the authors really have any other results (conference, workshop) to cite here?

As stated by the reviewer, another fuzzy logic based S-RVS method presented at The 8th European Congress on Computational Methods in Applied Sciences and Engineering ECCOMAS Congress in June 2022 has been explained along with the current reference. Both of the S-RVS methods in these references are developed by using the methodology presented in this study. (from lines 119 to 123)

  1. Line 131. "URM buildings" should be explained. What is it? Maybe refer to the Fig. 1.

Since the sentence, which contains the highlighted abbreviation (URM), already refers to Figure 1, no further explanation has been added. The examples given in this sentence are shown in Figure 1, which is the combination of three different figures corresponding to three examples in the sentence, respectively.

  1. Figure 1 and 11 copyright? by authors?

The second author of this study as part of the team dispatched to Albania by the Hungarian government collected building screening data and corresponding pictures following the 2019 Albania earthquake.

  1. Figure 4. Spectral acceleration values ... corresponding to the building height. Where is height? I see only period. What red dots mean?

Another sentence was added to the paragraph before Figure 4 to demonstrate the relation between the evaluated spectral acceleration values and the number of stories of buildings upon the reviewer's specification.

  1. Line 315 style for "Table 4."

Table 4's style was altered to MDPI format by restructuring.

  1. I do not like subsection 2.9. It has uncertain places.
  • Line 257 "Error! Reference source not found".

The error in the relevant area has been fixed by referring to Equation (1).

  • Lines 260&262 not Figure 4, but Figure 3!

Even though the reviewer asked that Figure 4's in-text references are required to be changed for refering to Figure 3, the authors' most recent examination revealed that the references provided are accurate.

  • For example, bullet point 2 mentions the "blue (continuous) line". Unlike Figure 3, the "blue (continuous) line" is shown in Figure 4.
  • For example, bullet point 3 mentions "red scatters". Unlike Figure 3, "red scatters" are shown in Figure 4.
  • Line 257 not just period, but "fundamental structural period".

As stated by the reviewer, changes have been made in the relevant part.

  • Please, try to save figure titles NEAR the figure.

The naming of Figure 3 and Figure 4 has been changed to be close to the relevant figures. In addition, the naming inside Figure 3 is modified to be near to the related Figure.

  1. Finally, I am lost. Authors call Ta with different names. Line 265 - building fundamental period ; Line 267 - approximate fundamental period; line 271 - fundamental building period; Figure 3 - fundamental structural period. Please, use the same term in the entire text.

The explanation of Ta has been modified to mention to the "fundamental structural period" in all relevant places, as the reviewer noted.

  1. OK, we see an equation in the figure 3. So, the repetition in the text is excessive.

The equation given in Figure 3 is removed to avoid repetition of equation 1. Since the equation and corresponding explanation were removed from Figure 3, further modification has been made in Equation 1 and its following explanation.

  1. Line 279 what is "FIS system", which was never mention before?

Since this study details the background research acquired to design a fuzzy logic-based S-RVS method FIS has been changed to the fuzzy logic system in the relevant parts.

  1. Ct dimension? If Ta in period is sec, h - height in meters. But upon Fig. 3 - the coefficient of building period is equal 0.05.

Ct is a constant created to establish a relationship between building height and fundamental structural period. Ct takes different values for different buildings such as for steel moment resisting frames 0.085, for concrete moment resisting frames 0.075 is assigned to Ct. As explained in Figure 3, Ct for URM structures is equal to 0.05. The corresponding reference [60] is provided in line 266.

  1. Figure 5. upon Zadeh [63]?

Figure 5 was created by the authors based on the information presented in the literature to offer the reader with a broad framework of fuzzy logic.

  1. Figure 7d. d) Building vulnerability (e) Site seismic hazard. In the figure, I observe "very low low moderate high very_high". However, only in Table 5 (below) I see, that only Building vulnerability and Site seismic hazard have 5 classes, and other - 3 classes. This needs more in the subsections of Section 2. Also are you sure "2.12. Building Damageability", so where do you explain Building vulnerability ?

Generally, the input parameters are discussed in Section 2 to help the implementer (such as an engineer or architect) comprehend the subject in greater depth. However, as mentioned in the paragraph before Figure 7, intermediate parameters are a form of product of the other parameters. Furthermore, a parameter may be defined by constructing different membership functions as necessary. For example, instead of low, moderate, and high, they may be very low, low, moderate, high, and very high. Because it is critical to classify structures based on the determined building damageability index, building damageability is provided after discussing input parameters under section 2.

  1. Line 395 too early to Figure 9. The mention in the text is in line 458.

Since Figure 9 is referred to in the text on line 394 (just before Figure 9), the location of the relevant figure has not been changed.

  1. All figures move right.

All of the tables and figures were moved to the right.

  1. Tables: inappropriate style. Please, refer to the table template.

Style of table 1, 3, 4, 5, 6, 7 changed to the MDPI format.

  1. In my opinion, authors need to acknowledge anonymous reviewers, as they find so many small mistakes. And reviewers help significantly in manuscript improving.

Since it is widely accepted that the mistakes in the manuscript should be reported to the authors by the reviewers, and it is accepted that the publication of the relevant article is possible thanks to the comments of the reviewers, anonymous referees were not acknowledged in the Acknowledge.

References.

  1. First of all, journal's titles - italic!

All journal titles were modified to be italics.

  1. For books and guidance - Publisher city and country,+Publisher. or online link.

All the modifications that have been performed to fulfill the comments of the reviewer are shown in green text in the References section. In addition link to the standards has been added.

  1. g. [10-14] - "." at the end not ";"

The corresponding places were modified as stated by the Reviewer.

  1. [67] initials! Mamdani E.H.

The corresponding reference was modified as stated by the reviewer.

The adjustments and additions made in this study are shown in green text in the paper, making it easy to follow all of the changes.
